# A functional SNP rs895819 on pre-miR-27a is associated with bipolar disorder by targeting NCAM1

Yifeng Yang [1,6✉], Wenwen Lu[2,6], Mei Ning[3,4,6], Xianhao Zhou[1,6], Xinyao Wan[1], Qianglong Mi[1], Xiaoyan Yang[2], Di Zhang[3,4], Yuanyuan Zhang[1], Biao Jiang[1], Lin He[3,4], Jia Liu [1,2,5✉] & Yan Zou [2✉]

The aberrant expression or genomic mutations of microRNA are associated with several human diseases. This study analyzes the relationship between genetic variations of miRNA and schizophrenia or bipolar disorder. We performed case-control studies for ten SNPs in a total sample of 1584 subjects. All these ten SNPs were on or near mature microRNAs. We identified the association between bipolar disorder and the T/C polymorphism at rs895819. To illustrate the function of miR-27a, we constructed several miR-27a knockout (KO) cell lines, determined candidates of miR-27a, and then verified *NCAM1* as a target gene of miR-27a. Further studies revealed that the T/C polymorphism on miR-27a led to the differential expression of mature and precursor miR-27a without affecting the expression of primary miR-27a. Furthermore, the C mutation on pre-miR-27a suppresses cell migration and dopamine expression levels. Our study highlights the importance of miR-27a and its polymorphism at rs895819 in bipolar disorder.

[1] Shanghai Institute for Advanced Immunochemical Studies, ShanghaiTech University, 393 Middle Huaxia Road, 201210 Shanghai, China. [2] School of Life Science and Technology, ShanghaiTech University, 201210 Shanghai, China. [3] Key laboratory for the Genetics of Developmental and Neuropsychiatric Disorders (Ministry of Education), Bio-X Institutes, Shanghai Jiao Tong University, 1954 HuaShan Road, Shanghai 200030, China. [4] Institute for Nutritional Sciences, Shanghai Institutes for Biological Sciences, Chinese Academy of Sciences, Shanghai, China. [5] Shanghai Clinical Research and Trial Center, Shanghai, China. [6] These authors contributed equally: Yifeng Yang, Wenwen Lu, Mei Ning, Xianhao Zhou. ✉email: yangyf1@shanghaitech.edu.cn; liujia@shanghaitech.edu.cn; zouyan@shanghaitech.edu.cn

Schizophrenia and bipolar disorder are two major types of psychiatric diseases. While there are significant efforts to analyze the genetic causes of psychiatry[1], most studies are complicated by environmental factors[2]. Along with independent effects, genetic and environmental factors can contribute to the occurrence and progression of psychiatric diseases[3]. For example, early life adversity can trigger genome-wide pathogenic changes in DNA methylation, which is a major risk factor for depressive disorder and schizophrenia[4]. A series of gene families can mediate the epigenetic response to environmental stresses, including non-coding microRNA (miRNA)[5].

MiRNAs are a class of single-stranded, evolutionarily conserved non-coding RNAs comprising approximately 22 nucleotides. MicroRNAs modulate gene expression at the post-transcriptional level by either repressing messenger RNA (mRNA) translation or selectively degrading mRNA[6]. Unlike mRNA, the maturation of miRNA requires both nuclear and cytoplasmic processing. MiRNA biogenesis begins with the transcription of primary miRNA (pri-miRNA) by RNA polymerase II in the nuclei. Pri-miRNA contains a 5′ cap and poly (A) tail and is typically several kilobases long, encoding one or several precursor miRNAs (pre-miRNA)[7]. DiGeorge syndrome, in the critical region gene 8 (DGCR8), recognizes the stem-loop (hairpin) structure of pri-miRNA and renders it to RNase III enzyme Drosha. Drosha cleaves on the stem of pri-miRNA, releasing the pre-miRNA, which is composed of approximately 70 nucleotides with a hairpin structure[8]. Pre-miRNA is then transferred into the cytoplasm and cleaved by another RNase III enzyme known as Dicer, producing the mature, double-stranded RNA duplex with 18−25 nucleotides[9]. Mature miRNA consists of an antisense or guide strand and a sense or passenger strand. The guide strand integrates with Argonaut (Ago) proteins to form the RNA-induced Silencing Complex (RISC)[10]. RISC can interact with the 3′ untranslated region (UTR) of a target mRNA via Watson-Crick pairing, leading to destabilization or translational repression of the target mRNA[11]. Many miRNAs are expressed in the human brain[12] and are known to play important roles in various psychiatric diseases, including depressive disorders[13], autism[14], anxiety, and stress-related disorders[15], bipolar disorder, and schizophrenia[16]. Importantly, miRNAs can act as epigenetic regulators in post-traumatic stress disorders[17,18]. Drug treatments of psychiatric diseases are often associated with altered miRNA expression[19,20]. These data strongly suggest a genetic association between miRNA and psychiatric diseases.

In this study, we analyzed the association between genetic variations of miRNA and schizophrenia or bipolar disorder in the Chinese Han population. A single nucleotide polymorphism (SNP) of T/C at rs895819 in miR-27a was found to be associated with bipolar disorder. Using a miR-27a KO cell model, we identified several candidate target genes of miR-27a and verified NCAM1 as a target gene of miR-27a by western blotting, qPCR, and luciferase reporter assay. Furthermore, we characterized the effects of T/C mutation on the expression of target genes and the maturation of miR-27a, thus influencing cell migration and dopamine secretion expression.

## Results

**Association study.** In this study, we collected samples from the Chinese Han population in southern China. Three groups were analyzed: schizophrenia, bipolar disorder, and healthy individuals (control). In total, there were 1584 subjects. We genotyped the SNPs within the 300 bp region of mature miRNA and identified 457 SNPs in or near 307 miRNAs. The ten SNPs on schizophrenia or bipolar disorder susceptible loci with minor allele frequency larger than 0.05 (Table 1) were further investigated. We found that rs895819 on pre-miR-27a was associated with the incidence of bipolar disorder ($p = 0.0197$) but not with that of schizophrenia (Table 2). In power calculations, we found that the sample size had >90% power for rs895819 to detect gene effect OR = 1.25 with $a \leq 0.05$. Therefore, we focused on rs895819 in subsequent analyses.

**Construction of miR-27a knockout astrocytoma cell line.** The altered cell number and activity of astrocytes are associated with bipolar disorder[21]. Therefore, in this study, we investigated the potential targets of miR-27a in astrocytes. We chose U-251MG astrocytoma cells as our model cell line and used a CRISPR/Cas9 system to generate miR-27a knockout U-251MG cells. A single guide RNA (sgRNA) targeted to the pre-miR-27a sequence was designed and delivered, along with Cas9 nuclease to U-251MG cells using lentivirus (LVs). DNA sequencing analyses indicated successful gene editing at the targeted genomic locus (Supplementary Fig. 1a). We isolated three single clones containing dual allelic gene modifications at the pre-miR-27a site (Supplementary Fig. 1b) and chose miR-27-KO in U251 (clone 1) and miR-27-KO in SH-SY5Y (clone 3) for further analyses.

**RNA-Seq analyses.** We performed whole transcriptome (RNA-Seq) analyses on three biological replicates of each wild-type (WT) and miR-27a knockout (KO) U-251MG cell. Cluster analyses of the sequenced samples suggested a significantly altered gene expression profile (Supplementary Fig. 2). In total, 681 significant differentially expressed genes (DEGs) were identified, consisting of 215 (1.4%) upregulated and 466 (3.0%) downregulated genes in miR-27a KO cells (Fig. 1a). Gene ontology (GO) analyses demonstrated that miR-27a KO notably enriched development-related GO terms in biological processes (BP) (Fig. 1b). For many of these GOs, enrichment was found in more than 20% of the genes within the same cluster (cluster frequency) (Fig. 1c), suggesting that miR-27a plays a critical role in the developmental process. Enriched GO terms in cellular components (CC) are primarily related to the plasma membrane or extracellular matrix. In line with the enriched CC GO terms, enriched molecular function (MF) GOs include many ligand-receptor binding-related terms (Fig. 1d).

**Identification of the target genes of miR-27a in U-251MG cells.** We used a strict 0.5 FPKM cutoff to generate a list of genes with significant base level expression and fewer false positives than a lower threshold would. To identify the potential target genes of miR-27a, we analyzed the DEGs which are predicted to harbor miR-27a seed sequence (Target Scan or miRDB) in WT and miR-

### Table 1 Identified SNPs in miRNA genes.

| rs number | miRNA | Chr | Position | Allele frequency (CHB) |
|---|---|---|---|---|
| rs543412 | hsa-miR-100 | 11 | 121528137 | C: T = 0.636:0.364 |
| rs12903401 | hsa-miR-184 | 15 | 77289151 | G: C = 0.411:0.589 |
| rs2304608 | hsa-miR-9-2 | 5 | 87998144 | G: T = 0.542:0.458 |
| rs629367 | hsa-let-7a-2 | 11 | 121522224 | C: A = 0.167:0.833 |
| rs2910164 | hsa-miR-146a | 5 | 159844996 | C: G = 0.444:0.556 |
| rs895819 | hsa-miR-27a | 19 | 13808292 | T: C = 0.689:0.311 |
| rs107822 | hsa-miR-219-1 | 6 | 33283553 | A: G = 0.567:0.433 |
| rs11614913 | hsa-miR-196a-2 | 12 | 52671866 | C: T = 0.489:0.511 |
| rs1011784 | hsa-miR-27b | 9 | 96887835 | C: G = 0.344:0.656 |
| rs7372209 | hsa-miR-26a-1 | 3 | 37985712 | T: C = 0.322:0.678 |

**Table 2 Allele-wise association analyses of SNP between psychiatry patients and healthy individuals.**

| Allele results in schizophrenia | | | miRNA | SNP | Allele | Allele results in bipolar disorder | | |
|---|---|---|---|---|---|---|---|---|
| OR | 95% CI | *P*-value[a] | | | | OR | 95% CI | *P* value[a] |
| 0.929 | 0.779–1.108 | 0.414 | hsa-miR-100 | rs543412 | C/T | 1.024 | 0.862–1.217 | 0.786 |
| 0.874 | 0.736–1.039 | 0.127 | hsa-miR-184 | rs12903401 | C/G | 1 | 0.846–1.185 | 0.999 |
| 1.165 | 0.981–1.385 | 0.082 | hsa-miR-9-2 | rs2304608 | A/C | 1.137 | 0.960–1.347 | 0.136 |
| 1.169 | 0.952–1.437 | 0.137 | hsa-let-7a-2 | rs629367 | A/C | 1.216 | 0.993–1.489 | 0.058 |
| 1.083 | 0.908–1.293 | 0.373 | hsa-miR-146a | rs2910164 | C/G | 1.044 | 0.879–1.240 | 0.625 |
| 1.118 | 0.918–1.361 | 0.268 | hsa-miR-27a | rs895819 | C/T | 1.253 | 1.037–1.515 | **0.0197**[b] |
| 0.932 | 0.778–1.116 | 0.444 | hsa-miR-219-1 | rs107822 | C/T | 0.982 | 0.824–1.171 | 0.842 |
| 0.986 | 0.829–1.173 | 0.87 | hsa-miR-196a-2 | rs11614913 | C/T | 1.065 | 0.899–1.261 | 0.466 |
| 1.002 | 0.839–1.197 | 0.979 | hsa-miR-27b | rs1011784 | C/G | 0.922 | 0.776–1.097 | 0.361 |
| 0.958 | 0.795–1.154 | 0.65 | hsa-miR-26a-1 | rs7372209 | C/T | 1.005 | 0.837–1.208 | 0.954 |

[a]*P* values calculated by χ2-test or Fisher's exact test.
[b]*P*-value <0.05 is in boldface, and the risk allele is C allele.

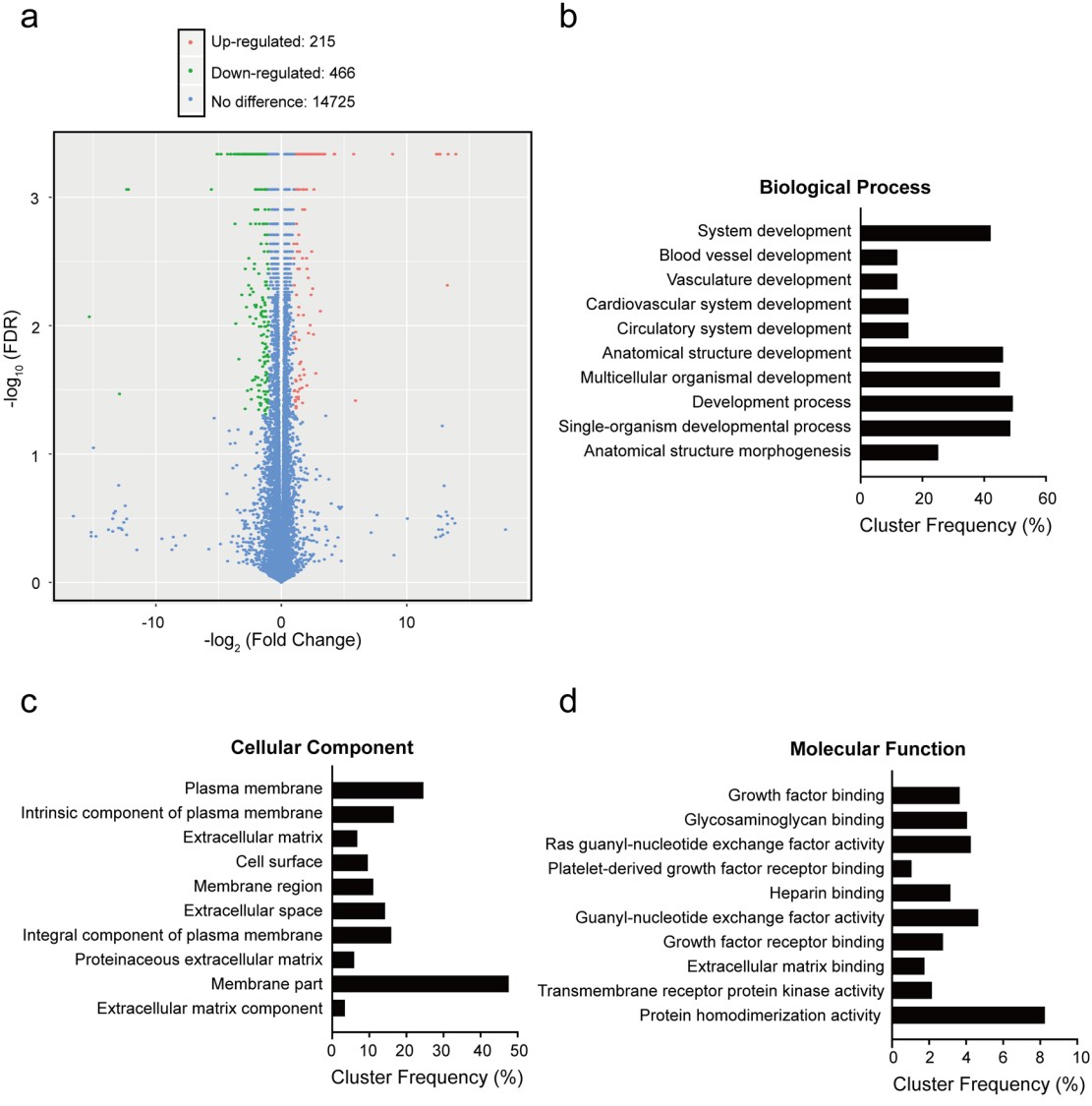

**Fig. 1 RNA-Seq analyses of the function of miR-27a. a** Volcano plot of significant differentially expressed genes (DEGs). **b** Gene ontology analyses showing top 10 most significantly enriched GO terms in biological process. **c** Top 10 most significantly enriched GO terms in cellular component. **d** Top 10 most significantly enriched GO terms in molecular function categories.

**Table 3 Top 20 upregulated genes of miR-27a candidates in RNA-Seq analyses.**

| Gene ID | Gene name | Log2(FKPM_KO/FKPM_WT)[a] | P value[b] |
|---|---|---|---|
| ENSG00000255690 | TRIL | inf | 5.00E−05 |
| ENSG00000090339 | ICAM1 | 4.20636 | 5.00E−05 |
| ENSG00000204941 | PSG5 | 3.41963 | 5.00E−05 |
| ENSG00000089486 | CDIP1 | 2.21503 | 5.00E−05 |
| ENSG00000242265 | PEG10 | 1.81353 | 5.00E−05 |
| ENSG00000064300 | NGFR | 1.63547 | 5.00E−05 |
| ENSG00000107796 | ACTA2 | 1.52824 | 5.00E−05 |
| ENSG00000160712 | IL6R | 1.35124 | 5.00E−05 |
| ENSG00000113070 | HBEGF | 1.31657 | 5.00E−05 |
| ENSG00000106683 | LIMK1 | 0.935051 | 5.00E−05 |
| ENSG00000151012 | SLC7A11 | 0.93043 | 5.00E−05 |
| ENSG00000122786 | CALD1 | 0.8746 | 5.00E−05 |
| ENSG00000137642 | SORL1 | 0.806905 | 5.00E−05 |
| ENSG00000109586 | GALNT7 | 0.803907 | 5.00E−05 |
| ENSG00000149294 | NCAM1 | 0.781666 | 5.00E−05 |
| ENSG00000064393 | HIPK2 | 0.764313 | 5.00E−05 |
| ENSG00000119042 | SATB2 | 0.744465 | 5.00E−05 |
| ENSG00000119771 | KLHL29 | 0.722311 | 5.00E−05 |
| ENSG00000184371 | CSF1 | 0.5667 | 5.00E−05 |
| ENSG00000146648 | EGFR | 0.4306 | 5.00E−05 |

[a]FKPM > 0.5 at least in one group was used to generate a list of genes with significant base level expression and fewer false positives than a lower threshold would.
[b]Genes have been filtered to have an adjusted P value of less than 0.001.

27a KO U-251MG cells in the RNA-Seq analyses and determined the top 20 genes with the most upregulated expression in the KO cells (Table 3). We then validated the mRNA expression of these genes in WT and miR-27a KO cells using real-time quantitative PCR (RT-qPCR) in two cell lines. Of the top 20 genes screened by RNASeq, most were upregulated in both the U251 KO mutant and the SH-SY5Y KO mutant, meaning that the expression level of these genes in the knockout mutant exceeds that of the wild-type. Six genes (NCAM1, PEG10, ICAM1, IL6R, LIMK1, and HIPK2) were found to have expression levels 1.5-fold that were higher in miR-27a KO cells (P < 0.001, Fig. 2). This suggests that these six genes could be potential target genes of miR-27a in U-251MG and SH-SY5Y cells.

**The effects of rs895819 polymorphism on miR-27a expression.** Rs895819 is located at the terminal loop of pre-miR-27a[22], meaning that the T to C mutation could affect the maturation process of miR-27a. Therefore, we determined the expression level of mature miR-27a in NPCs. Mutant miR-27a exhibited significantly lower expression levels (p < 0.001, Fig. 3a), indicating that the maturation process or the stability of miR-27a could be affected by rs895819 variation (T to C). To further elucidate the mechanism underlying these decreased miR-27a levels, we analyzed the expression of pri- and pre-miR-27a. Interestingly, pri-miR-27a-T and pri-miR-27a-C had similar expression levels in NPCs whereas pre-miR-27a-C had significantly reduced expression levels compared to pre-miR-27a-T (p = 0.0123, Fig. 3b). These results indicate that the T to C mutation at rs895819 affected the maturation of miR-27a from the primary form to the precursor form. The reduced expression of mature and precursor microRNA with the mutant miR-27a-C can explain the impaired inhibitory effects of miR-27a-C on the target genes.

**ICAM1, NCAM1 as candidate targets of miR-27a.** Of the six genes showing 1.5-fold more expression in miR-27a KO cells, ICAM1[23] and NCAM1[24] were reported to be candidate genes for both bipolar disorder and schizophrenia. The relative expression of mature miR-27a was dramatically increased by the miR-27a mimic and decreased by the miR-27a inhibitor or miR-27 KO

(p < 0.0001) in U251 and NPC (Fig. 4a, d), which demonstrates the effectiveness of miR-27a mimics, inhibitors, and knockout. We found one isoform of NCAM1 at 60KDa, which was significantly decreased by the miR-27a mimic and increased by the miR-27a inhibitor (p < 0.05) in NPC (Fig. 4b and Supplementary Figs. 3, 4) The main isoform of NCAM1 at 180KDa was also increased by the miR-27a inhibitor (p < 0.05) but not decreased by miR-27a mimics in U251 (Fig. 4b and Supplementary Fig. 5). Meanwhile, ICAM1 is significantly downregulated by miR-27a mimics (p < 0.001) and upregulated by miR-27a inhibitors (p < 0.05, Fig. 4c, Supplementary Fig. 6). However, this was not as effective in NPCs (Fig. 4c and Supplementary Figs. 7, 8).

To minimize off-target CRISPR, we constructed three miR-27 KOs in U251 using three different sgRNAs (sg30-mix, sg31-mix, sg32-mix, Supplementary Fig. 9). Although the mRNA expression of NCAM1 and ICAM1 did not show similar trends to that of protein levels (Fig. 4e), we found that the protein expression of ICAM1 and NCAM1 showed obvious increases in miR-27a KO cell lines compared to the U251 wild type (p < 0.001, Fig. 4f, g, Supplementary Figs. 10, 11). Therefore, ICAM1 and NCAM1 were suggested as candidate targets of miR-27a.

**Identifying NCAM1 as a direct target as miR-27a.** ICAM1 and NCAM1 were predicted as the targets of miR-27a using Targetscan (Fig. 5a). To examine the direct or indirect relationship between miR-27a and ICAM1 & NCAM1, we utilized a 3′UTR luciferase reporter assay. We found that relative luciferase activity of 3′UTR-NCAM1 decreased dramatically in cells transfected with miR-27 mimic compared to the miR-27 NC group, while upregulation of miR-27a had almost no effect on the relative luciferase activity of NCAM1-Mutant (Fig. 5b). In addition, we did not detect the significant changes of relative luciferase activity on 3′UTR-ICAM1 (Fig. 5b). This indicates that NCAM1 but not ICAM1 is a direct target of miR-27a.

**MiR-27a-C mutant suppresses cell migration and inhibits the secretion level of dopamine transmitter.** To evaluate the effects of rs895819 mutation on the neuron, we constructed mammalian expression plasmids containing either WT (T) or mutant (C)

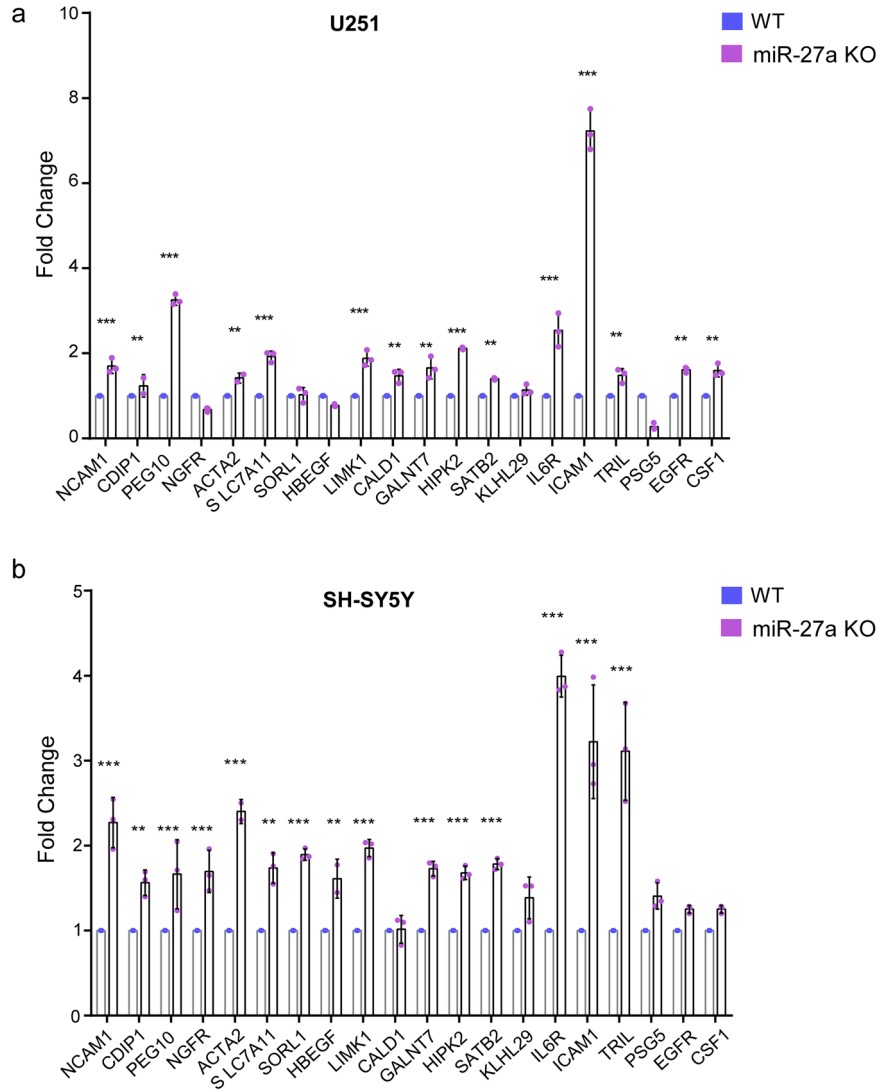

**Fig. 2 RT-qPCR validation of upregulated genes in miR-27a knockout (KO). a** RT-qPCR validation of upregulated genes in miR-27a knockout (KO) U251 cells compared to wild type (WT). **b** RT-qPCR validation of upregulated genes in miR-27a knockout (KO) SH-SY5Y cells compared to wild type (WT). Significant differences between WT and KO were analyzed by multiple unpaired t-tests; $n = 3$, adjusted $p$-values using Holm–Šídák method. Data were shown as mean ± SD. (*, $p < 0.05$; **, $p < 0.01$; ***, $p < 0.001$).

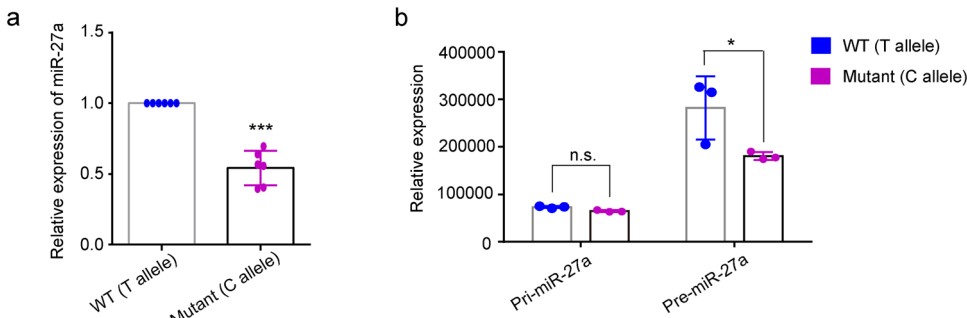

**Fig. 3 Expression of WT and mutant miR-27a in NPCs. a** Relative expression of mature miR-27a in NPCs. U6 snRNA is used as an internal reference. Significant difference between WT and mutant was analyzed using unpaired Student's t-test (***, $p < 0.001$). **b** Relative expression of pri- and pre- miR-27a in NPCs. *GAPDH* is used as an internal reference. Significant difference between WT and mutant was analyzed using two-way ANOVA (*, $p < 0.05$, $p = 0.0123$; n.s., not significant). *P* value is reported to one of the main effects for two-way ANOVA statistics. Data shown as mean ± SD of at least three replicates.

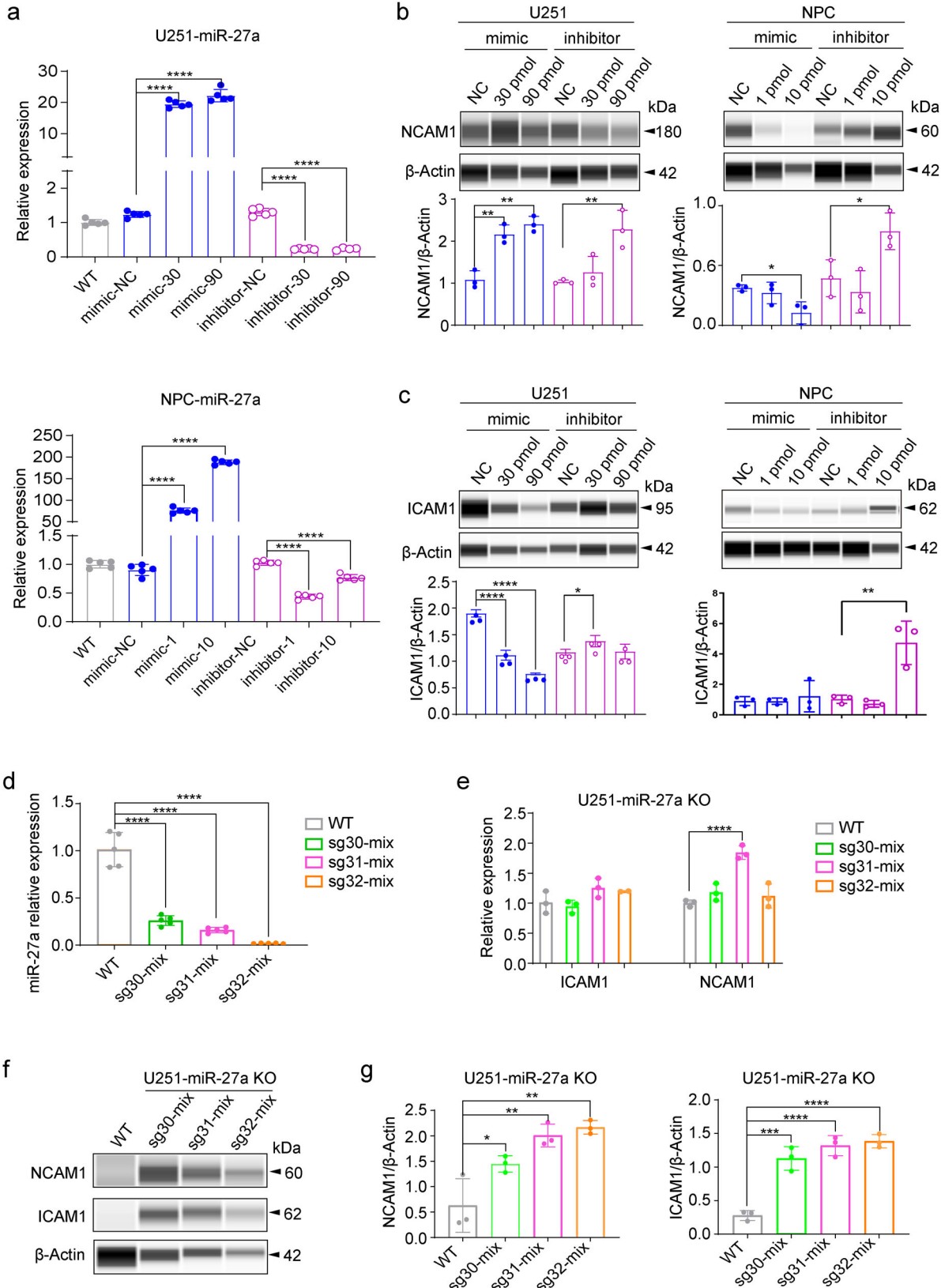

miR-27a sequences (miR-27a-T and miR-27a-C) for over-expression. These experiments were performed in neural progenitor cells (NPCs) or SH-SY5Y.

After 21 days of induction, the NPC cells were induced into neurons (Fig. 6a). We found the dopamine transmitter was reduced in the miR-27a-C-Mutant (Fig. 6b, $p < 0.01$), but the GABA was unchanged (Fig. 6c). Furthermore, cell migration was inhibited in the miR-27a-C-Mutant in SH-SY5Y cell line (Fig. 6d, e, $p < 0.01$). This is consistent with the dysregulation of the dopamine hypothesis in bipolar disorder.

**Fig. 4 ICAM1 and NCAM1 are candidate targets of miR-27a in U251 and NPC. a** The relative expression of miR-27a transfected by miR-27a mimics and inhibitor in U251 and NPC were determined by qRT-PCR. The two doses in U251 were 30 pmol and 90 pmol and in NPC were 1 pmol and 10 pmol respectively, with five replicates ($n = 5$) for every sample. **b, c** Show the relative protein expression levels of NCAM1 and ICAM1 transfected by miR-27a mimics and inhibitor in U251 and NPC, and its quantification using Compass Simple Western software. Date are representatives with three replicates ($n = 3$) for each sample. Significant differences between these groups were analyzed using one-way ANOVA, Tukey's multiple comparisons test for adjusted $P$ values. **d, e** The relative mRNA expression levels of mature miR-27a, NCAM1, and ICAM1 in miR-27 KO using three different sgRNAs compared to WT. Sg30-mix, sg31-mix, and sg32-mix denote the miR-27 KO in U251, $n = 5$. **f** The western blot of NCAM1 and ICAM1 in miR-27 KO using three different sgRNAs and WT, $n = 3$. **g** The quantification of NCAM1 and ICAM1 protein levels in U251 cells. Significant differences between these groups were analyzed using one-way ANOVA, and Tukey's multiple comparisons test were used for adjusted P values. Data were shown as mean ± SD. (*, $p < 0.05$; **, $p < 0.01$; ***, $p < 0.001$; ****, $p < 0.0001$).

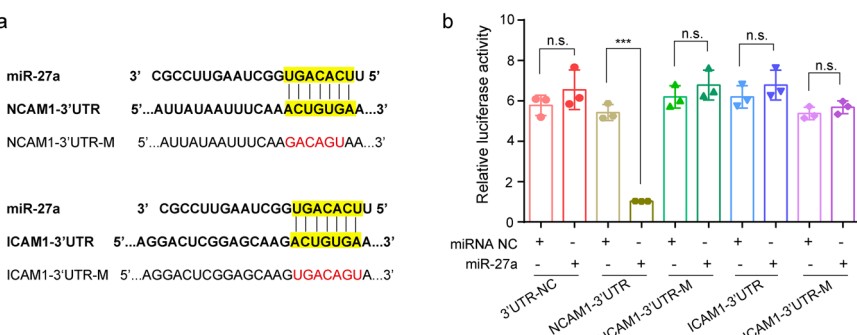

**Fig. 5 NCAM1 is a direct target of miR-27a. a** The miR-27a binding sites in the 3′UTR of ICAM1, NCAM1. The binding sites were predicted by Targetscan (marked in yellow). The mutated seed sequences of NCAM1-3′UTR and ICAM1-3′UTR were denoted in red. **b** 293T cells were co-transfected with pmiR-3′UTR-WT/pmiR-3′UTR-M and miR-27 mimic/inhibitor or its NC. Luciferase activity was detected using a dual-luciferase assay system after 48 h. Unpaired t-test was performed between two groups with five replicates ($n = 5$), * indicates the significant difference compared with NC group (***, $p < 0.001$; n.s, not significant). Data were shown as mean ± SD.

## Discussion

MiR-27a is associated with a broad range of human diseases. Aberrant expression or genetic mutations of miR-27a can lead to colorectal cancer (CRC)[25,26], breast cancer[27–29], ovarian cancer[30], gastric cancer[31,32], and other cancers. More importantly, SNPs of miR-27a at rs895819 are associated with a high risk of CRC[33] and non-small-cell lung cancer (NSCLC)[34] in the Chinese Han population. Hsa-miR-27a is considered an oncogene, and its expression level is abnormal in many types of cancers. Interestingly, the C allele of rs895819 in our study was identified as a risk allele for bipolar disease, but in cancer, it was a protected allele[35]. This phenomenon could reflect differences in evolutionary adaptation, which could cause opposite effects in different diseases.

In the current study, we analyzed the association between genetic variations in microRNA and two psychiatric diseases (bipolar disorder and schizophrenia) in the Chinese Han population. We found that the T to C mutation at rs895819 of miR-27a was associated with bipolar disorder, but not with schizophrenia. In previous studies, schizophrenia and bipolar disorder have been grouped together, though they have many differences, from phenotype to etiology. Franks et al. found susceptible loci in chromosome 19q13 for bipolar but not for schizophrenia[36]. Moreover, Izumi et al. observed a significant allelic association between several SNPs and bipolar disorder but not for schizophrenia or depression[37], which aligned with our results. We believe that genetic differences between bipolar disorder and schizophrenia can be clinically useful for distinguishing these two psychiatric diseases and could provide help for specific treatment of different based on genetics.

MiR-27a may have different target genes in different cell lines. Our functional experiments of mimic and inhibitor indicate that in NPC cell line the target gene of miR-27a is NCAM1. CRISPR/Cas9-mediated knockout of all miR-27/24 in ESCs leads to serious deficiency in ESC differentiation in vitro and in vivo[38]. In our study, we found nucleotide polymorphisms (SNPs) or mutations occurring in the miRNA gene region may affect the property of miRNAs through altering miRNA expression and/or maturation. However, the mutation C on pre-miR-27a did not affect the cell growth rate in NPC (Supplementary Fig. 12). On the other hand, KO of miR-27a in NPCs affects their proliferation while cell growth is also inhibited in miR-27a KO of SH-SY5Y (Supplementary Fig. 13). This indicates that miR-27 also plays an important role in the proliferation and differentiation of NPC or neuron-like cell, and its function can be further studied in future experiments.

To illustrate the role of rs895819 on pre-miR-27a in bipolar disorder, we combined RNA seq, luciferase reporter assay, and downstream functional experiments to identify NCAM1 as a target gene of miR-27a in NPC. The neural cell adhesion molecule (NCAM) belongs to the immunoglobulin superfamily and is a cell recognition molecule bound to the membrane. It is involved in the development of the nervous system by regulating synaptic plasticity, neurite outgrowth, neuronal migration, and synapse formation[39]. NCAM1 is reported to have many isoforms, and three main isoforms consist of 120, 14, and 180 KDa. NCAM1 isoforms, are dysregulated in neuropsychiatric disorders including bipolar disorder in the brain and cerebrospinal fluid[40]. We found one isoform of NCAM1 at 60 KDa. This short isoform of NCAM1 was dramatically elevated when miR-27a was reduced or knocked out. Additionally, SNPs found in the NCAM gene, the abnormal proteolysis or polysialylation of the NCAM protein can change its role in diseases related to neuropsychiatry[41]. Recent evidence has showed that upregulation of NCAM1 suppresses ameloblastoma cell migration[42], which is in accord with our findings.

The normal release of neurotransmitters plays an important role in maintaining the normal physiological functions of the

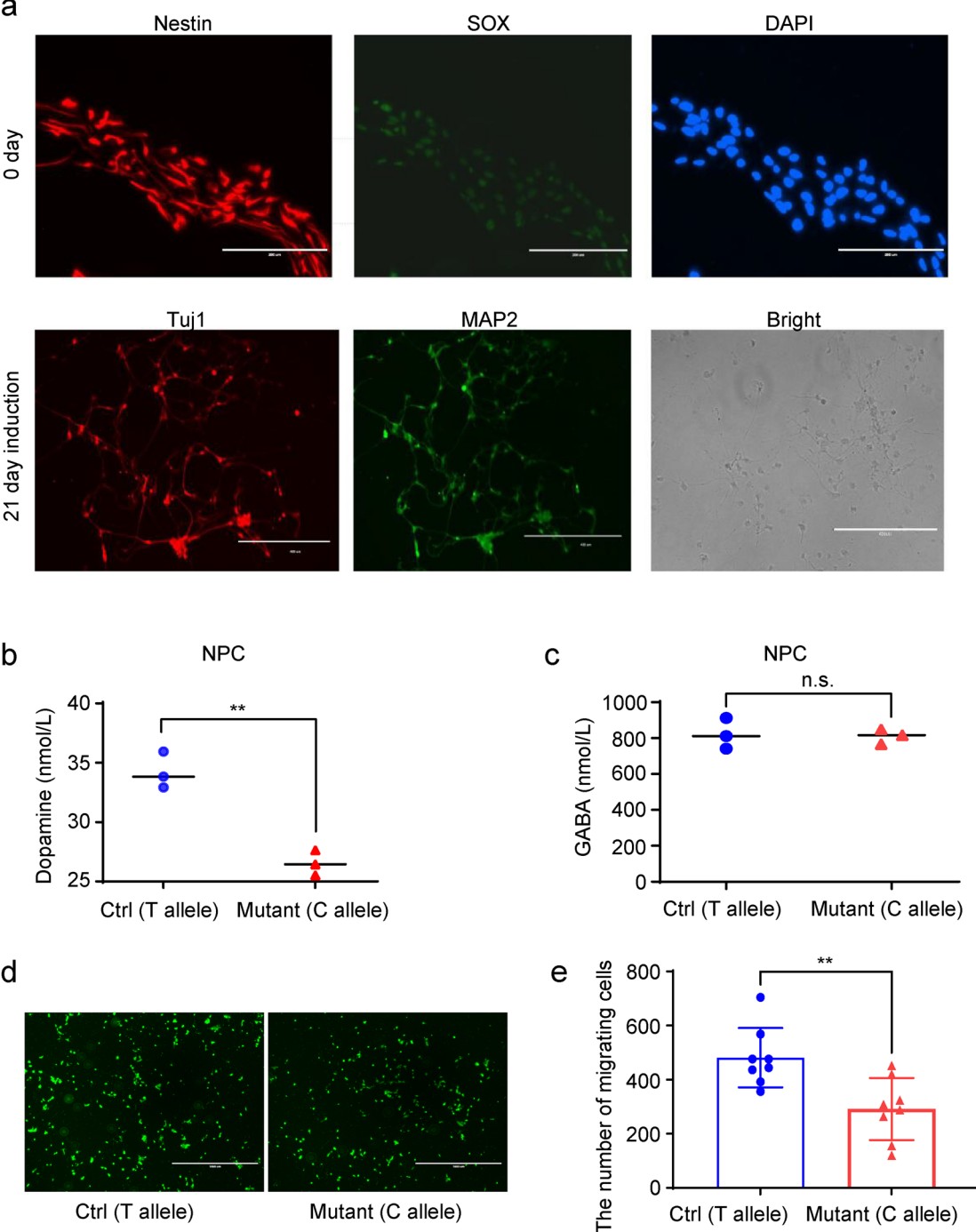

**Fig. 6 MiR-27a-C mutant suppresses cell migration and inhibits secretion levels of the dopamine transmitter. a** After 21 days of induction, the NPC cells (SOX[+] Nestin[+]) differentiated into neurons (MAP2[+]Tuj1[+]), scale bar 400 μm; **b**, **c** the expression of Dopamine and GABA by ELISA with three replicates ($n = 3$); **d** the image of transwell assay for miR-27a-Ctrl (T allele) and miR-27a-Mutant (C allele) using microscope; scale bar, 1000 μm. **e** the number of migrating cells with eight views for each group ($n = 8$). Unpaired t-test was performed in **b**, **c**, **e** between two groups; * indicates the significant difference between C allele and T allele (**, $p < 0.01$; *, $p < 0.05$; n.s, not significant, $p > 0.05$). Data were shown as mean ± SD.

body. Abnormalities in the release and regulation of neurotransmitters are also related to a series of pathological processes, such as depression and bipolar disorder[43]. There are hundreds of neurotransmitters in the human brain, including both excitatory neurotransmitters and inhibitory neurotransmitters. Dopamine[44] (one of the excitatory neurotransmitters) and GABA[45] (one of the inhibitory neurotransmitters) have been reported to be involved in the etiology of bipolar disorder. Our study found that the miR-27a-C mutant suppresses cell migration and inhibits

secretion levels of the dopamine transmitter. Western blotting of a miR-27a KO cell model and function analysis validated NCAM1 as a direct target in human NPCs. While NPCs[46] play critical roles in bipolar disorder, aberrant neuron functions might also be related to bipolar disorder[47]. Previous studies have identified associations of serum microRNA-27a with cognitive function in a Japanese population[48] and found that NCAM is elevated in the CSF of patients with bipolar type I and unipolar affective disorders[49].

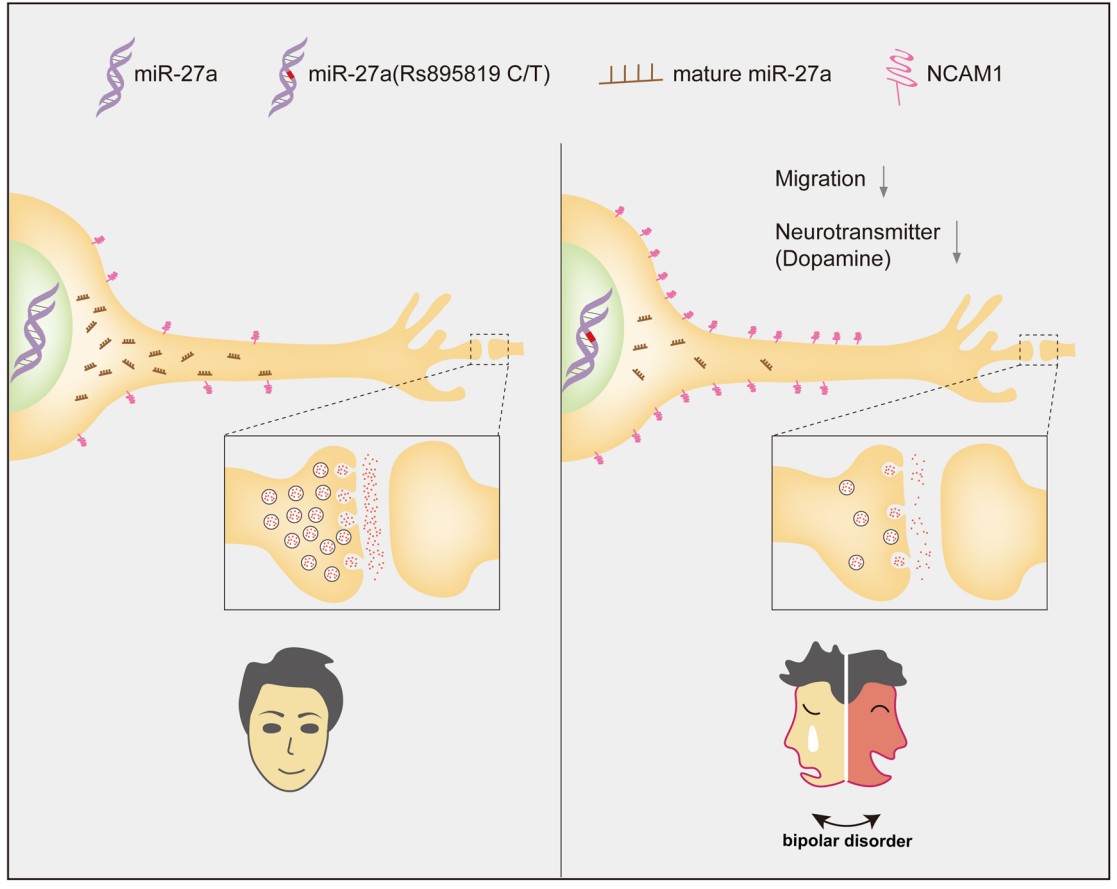

**Fig. 7 Possible mechanism of a functional SNP rs895819 on pre-miR-27a associated with bipolar disorder by targeting NCAM1.** The microRNA located in the intron of the chromosome genome is transcribed by RNA polymerase II to produce mRNA, and the initial microRNA is produced by RNA shearing. The initial transcription miRNA (PriRNA) is then cleaved by Drosha to produce the miRNA precursor (pre-miR-27a), where rs895819 (C/T) is located in the stem-loop region of pre-miR-27a, and finally digested by Dicer to form mature miR-27a. miR-27a-T had significantly higher expression than miR-27a-C in NPCs. The T to C mutation at rs895819 affected the maturation of miR-27a from primary form to precursor form. The expression of miR-27a was inhibited and elevated its target gene expression of NCAM1. It is hypothesized that upregulation of NCAM1 might suppress the cell migration and dopamine expression levels and thus result in the bipolar disorder.

All these indicate that miR-27a and its target NCAM1 may play an important role in bipolar disorder. Therefore, in future studies, it may be important to validate the identified target genes of miR-27a in primary or derived neurons. This all provides evidence for our hypothesis that miR-27a is involved in the etiology of bipolar disorder by targeting *NCAM1* (Fig. 7).

The current study has some limitations. First, we did not correct the number of tests performed in the population study, potentially increasing the risk of a false positive. Additionally, we did not successfully construct the KO model of miR-27a in NPC or neuron instead of the glioblastome cell line U-251MG. Future studies should examine the effects of the rs895819 mutation (T to C) on the interactions between miR-27a and DGCR8 or Drosha.

To the best of our knowledge, our study first revealed the potential role of miR-27a rs895819 in bipolar disorder. Although SNPs at rs895819 have been reported in the Chinese Han population, most of the existing studies are related to A/G polymorphism. Nevertheless, in this study, we identified T/C polymorphism at rs895819 and discovered the association between miR-27a and bipolar disorder by targeting NCAM1. Taken together, our results highlighted the critical role of rs895819 polymorphism for the expression and function of miR-27a, which also suggested a critical role of miR-27a in bipolar disorder.

## Methods

**Study population.** We recruited 1584 patients from southern China, including 528 schizophrenia patients with a mean age of onset at 47.3 ± 13.4 (52.3% male), 528 bipolar disorder patients with a mean age of onset at 41.1 ± 12.8 (55.3% male) and 528 unrelated healthy individuals with a mean age of 42.72 ± 13.1 (43.6% male) as the control group. Schizophrenia and bipolar disorder patients were diagnosed based on DSM-III-R criteria[50]. Each patient was assessed by at least two psychiatrists independently according to the case records and interviews. A standard informed consent was signed by each participant and reviewed and approved by the Shanghai Ethical Committee of Human Genetic Resources.

**SNP selection.** Since SNPs on or near the pre-miRNAs are more likely to influence the generation and function of mature miRNAs, we chose to analyze the SNPs within the 300 bp sequence of mature miRNA. Candidate miRNA genes were identified from three different ways: (1) the databases of miRBase version 18.0 and Hapmap Phase III were downloaded; according to miRBase version 18.0 data, the genomic location of miRNA genes was determined; by searching SNPs in or near miRNA genomic region according to Hapmap Phase III data then, we found 457 SNPs on or near (within 300 bp) 307 miRNA genes; (2) we searched in PUBMED database by using the keywords "microRNA", "brain", "CNS", "bipolar", "schizophrenia", "psychiatry" to find miRNAs which had been reported to be expressed in the brain and may play functional roles in psychiatric diseases. Finally, we chose 10 SNPs located in psychiatry susceptible loci, whose minor allele frequency was larger than 0.05 in Chinese Han population (Table 1).

**Genotyping and power calculation.** Genomic DNA was extracted from peripheral blood lymphocytes using phenol-chloroform method. Selected SNPs were genotyped by TaqMan® SNP Genotyping Assays (Applied Biosystems, Foster City, CA) on ABI PRIM 7900 Sequence Detection Systems. The results were then analyzed by SDS 2.2 software (Applied Biosystems) for allelic discrimination. Allele deviations

were assessed using Hardy Weinberg equilibrium (HWE) and the differences of allele and genotype frequencies between cases and controls were compared using SHEsis[51]. Three models were constructed using R program. Homozygote (1/1) and heterozygote (1/0) risk allele were coded as 2 and 1, respectively and homozygote non-pathogenic allele (0/0) were coded as 0. The dominant model was defined as $1/1 + 1/0$ versus $0/0$ and the recessive model as $1/1$ versus $1/0 + 0/0$. Power calculations were post-hoc calculations performed on the G*Power 3.0 program[52]. All reported $P$-values were two-tailed and statistical significance was defined as $p < 0.05$.

**Plasmids and cells.** LentiCRISPRv2 backbone plasmid was purchased from Addgene (#52961). The sgRNA targeted to pre-miR-27a was designed (gtggctaagttccgcccccc) and cloned into LentiCRISPRv2 using Esp3I (Thermofisher Scientific, Waltham, CA, USA) digestion as described[53] and this plasmid was referred to as lenti-CRISPR-miR-27a thereafter. For overexpression experiments, pri-miR-27a sequences containing T (wild type) and C (mutant) at rs895819 were cloned into pcDNA 3.1(+) under the control of CMV promoter. U-251MG cell line was purchased from the BeNa Culture Collection and cultured in DMEM (Thermofisher Scientific) Supplemented with 10% FBS (Thermofisher Scientific), 100 IU/mL of penicillin, and 100 mg/mL of streptomycin at 37 °C in a fully humidified atmosphere containing 5% CO2. U-251MG cells were verified prior to use by Saily Bio (Shanghai, China). Neural progenitor cells (NPCs) were purchased from ATCC (ACS-5003) and cultured in STEMdiff neural progenitor medium (STEMCELL Technologies, BC, Canada) at 37 °C in a fully humidified atmosphere containing 5% $CO_2$. The cell lines were tested for no mycoplasma contamination and authenticated at VivaCell Shanghai using short tandem repeat analysis.

**Cell line construction.** Lentivirus carrying Cas9 and sgRNA was produced by transfecting 293T cells with lenti-CRISPR-miR-27a plasmid along with two helper plasmids pMd2.G (Addgene #12259) and psPAX2 (Addgene #12260) using lipofectamine 3000 (Invitrogen, CA, USA). At 72 h post transfection, the supernatant containing the lentivirus was harvested, filtered, and stored at −80 °C for further application. U-251MG cells were infected with a MOI of 0.3 and selected using puromycin as described[54]. At 7 days after puromycin selection, gene modification was verified by DNA sequencing. Single clones carrying edited miR-27a sequence were isolated by limited dilution and genotyped by DNA sequencing. The primers for the PCR amplification of miR-27a are listed in Supplementary Table 1.

Overexpression stable line of miR-27a in SH-SY5Y and NPC were constructed using pCDH-Puro plasmid along with two helper plasmids as above and with 7-day puromycin selection.

**RNA-Seq analyses.** WT and miR-27a knockout U-251MG cells were harvested with three biological replicates in each group. Total RNA was extracted using Trizol reagent (Invitrogen, CA, USA). Whole-transcriptome sequencing was performed and analyzed by Vazyme Biotech (Nanjing, Jiangsu, China). RNA-Seq short reads were aligned to the human genome (GRCh38) using HISAT[55] with a maximum of two mismatches. On average, approximately 55.5 million reads across all samples were aligned to the reference, accounting for 96.5% of the total reads. Gene expression was counted as the number of short reads fully or partially aligned to the annotated gene model and was presented as expected number of Fragments Per Kilobase of transcript sequence per Million's base pairs sequenced (FPKM). Expressed genes were defined as genes with more than ten reads in total mapped in all samples and at least two of three replicates having more than two reads each. In total, 15,406 genes met the criteria and were defined as expressed in both WT and miR-27a KO U-251MG libraries. DEGs were identified using the CuffDiff module in CuffLink program[56] and are normalized to the library size between samples. $P$-values were adjusted for multiple testing using false discovery rates[57]. Significant DEGs were identified with an FDR ≤ 0.05 and a log2(fold change) ≥1. GO enrichment analyses were performed using Gorilla[58] by comparing the up/down-regulated DEGs to a list of all expressed genes. Significant GO terms with FDR ≤ 0.05 were reported.

**RT-qPCR analyses of candidate miR-27a targets.** The total RNA of WT and miR-27a KO U-251MG was extracted using Trizol reagent (Invitrogen, CA, USA) according to the manufacturer's instructions. One microgram of RNA was transcribed into cDNA using random primers and the SuperScript™ III First-Strand Synthesis System (Invitrogen, Massachusetts, USA). RT-qPCR was performed using the resulting cDNA templates and the 2 × SYBR Green qPCR Mix (Shangdong Sparkjade Biotechnology Co., Ltd.) in an Applied Biosystems 7900 Real-Time PCR Cycler (Applied Biosystems). Primers for RT-qPCR were designed by Beacon Designer 7.01 (www.premierbiosoft.com) and were listed in Supplementary Table 1. The reaction mix was denatured at 95 °C for 3 min and then subjected to 40 amplification cycles (10 s denaturation at 95 °C, 20 s annealing at 60 °C and 30 s extension at 72 °C) in a total of 20 µl system. The RT-qPCR data were analyzed using SDS software (Applied Biosystems). *GAPDH* or *U6* small nuclear RNA (snRNA) were used as internal control for mRNA and microRNA, respectively.

**Cell transfection.** MiR-27a mimic/inhibitor and their negative control (NC) were all purchased from Shanghai GenePharma Co. Ltd (Shanghai, China). We utilized Lipofectamine RNAiMAX reagent (Invitrogen, USA) to transfect miR-27a mimic/inhibitor or their corresponding control according to the protocol provided by the manufacturer. MiR-27a mimic and inhibitor at two doses (30 pmol, 90 pmol in U251 and 1 pmol, 10 pmol in NPC respectively) were used to upregulate or downregulate miR-27a expression. Then we investigated the relative mRNA expression of candidate target genes compared with negative control using qPCR technology.

**Protein quantification.** The miR-27a knockout mutant constructed on the U-251MG cell line and wild-type were plated in six-well plates at a density of $3 \times 10^5$ cells/mL. U-251MG and NPC were treated with miR-27a mimics and inhibitor using RNAiMAX at the indicated concentrations. Cells were collected after 72 h, washed with cold 1× PBS, and lysed in RIPA Lysis buffer (E121, GenStar) containing protease inhibitor cocktails (HY-K0010, MCE). Protein in cell lysate was quantified by detergent compatible Bradford assay kit (#23246, Thermo). Primary antibodies used in this study include ICAM1 antibody (ab53013, Abcam, 1:50 dilution), NCAM1 antibody (14255-1-AP, ProteinTech, 1:50 dilution) and β-actin (#4970, CST, 1:500 dilution). The protein level of ICAM1, NCAM1 and β-actin were evaluated by capillary western blot analyser Jess (ProteinSimple, Biotechne). We used 25-lane plates (12–230 kDa, 8 × 25 Capillary Cartridges) for protein detection according to the manufacturer's recommendations. Compass for SW (Simple Western, version 4.0) was used to the quantification of protein levels.

**Luciferase reporter assay.** 293T Cells were inoculated into 24-well plate and cultured to 80% confluence. The 3′-UTR of ICAM1/NCAM1 containing the wide type binding sites with miR-27 and the 3′-UTR of ICAM1/NCAM1 containing mutated binding sites were synthesized and sub-cloned into pmiR-report vector (Promega, Madison, WI, USA). Cells were co-transfected with pmiR-3′UTR-WT/pmiR-3′UTR-Mut and miR-27 mimic/inhibitor or its NC at a final concentration of 100 nM using Lipofectamine™ 3000 (Invitrogen, USA). After 48 h, the whole cell lysate was collected and the luciferase activity was then detected using a dual-luciferase assay system (E1910, Promega).

**The Induction of NPCs and neurotransmitter detection.** The NPCs were counted and seeded on to 6 cm dish to reach a confluency of 80% on the day of passage. The NPCs were detached by ACCUTASE (STEMCELL Technologies) and collected by centrifugation at 1000 rpm for 5 min and then seeded to coated 12 wells (5% matrigel, Biocoat) at a density of $4 \times 10^4$ cells/cm$^2$ culture in STEMdiff forebrain Neuron Differentiation medium (#08600, STEMCELL) with full medium change daily for 7 days and then culture in STEMdiff™ Forebrain Neuron Maturation media (#08605, STEMCELL) with full medium change every 2–3 days for 14 days.

After a total of 21-day induction, we collected the cell culture media of two kinds of neurons (induction individually from NPC-C-allele and NPC-T-allele) and then detected the secretion levels of neurotransmitters including dopamine and GABA using ELISA method (HB1693-Hu, HB2572-Hu, Shanghai Hengyuan company).

**Migration assay.** Transwell chamber with 8 µm pore size polycarbonate filter inserts for 24-well plates (Corning Costar Corporation, Cambridge, MA) were used to examine the migration of SH-SY5Y cells. We used a layer of membrane (Matrigel, Biocoat, 354480) to separate the high-nutrient culture solution from the low-nutrient culture solution. First, 100 µl 5% Matrigel (diluted by DMEM) was spread in the upper chamber and incubated at 37 °C for 2 h. Then we used the tips to suck off the excess Matrigel and washed it again with PBS. Next, $1 \times 10^4$ cells were seeded into the upper chamber separately with 200 µl serum-free DMEM. At the same time, 500 µl DMEM containing 20% FBS was added into the lower chamber. Briefly, serum-free DMEM containing cells were seeded into the upper chamber separately. The lower chamber was filled with culture supernatant of which acted as a chemotactic factor. After incubation for 48 h, the cells on the lower surface of the membrane were stained with stained with Calcium Green™ (1:100 dilution, C3011MP, AM) and imaged using an inverted microscope.

**Immunofluorescence.** Cells were seeded into a 12-well plate with glass cover slips were cultured overnight. The cells were then fixed with 4% paraformaldehyde, permeabilized with 0.5% Triton X-100, blocked with 3% bovine serum albumin, and incubated with primary antibody (Nestin, #4760 from CST, Dilution 1:200; Sox, SC-365823 from Santa Cruz, Dilution 1:200; Tuj1, GTX631836 from GeneTex, Dilution 1:500; MAP2, 17490-1-AP from Proteintech, Dilution 1:200) at 4 °C overnight. Secondary antibodies conjugated with Alexa fluorophore 488 or 647 were purchased from Invitrogen. Immunofluorescence was observed using a confocal microscope (Leica, Germany) and analyzed by Image J 1.4 software (Media Cybernetics, SilverSpring, MD).

**Statistics and reproducibility.** All experimental data were presented as means± standard deviation (S.D.). Unpaired two-tailed Student's t tests (T-test) were used to evaluate differences between two groups. One-way ANNOVA tests were used to

evaluate the significance of differences among three or more group. *P*-values were adjusted by Tukey's multiple comparisons test. All data were representative of at least three independent experiments and significant differences were indicated as *, $p < 0.05$; **, $p < 0.01$; ***, $p < 0.001$; ****, $p < 0.0001$. Two-way ANNOVA tests were performed in Fig. 3b. All calculations were performed using GraphPad Prism 9.0.

**Reporting summary**. Further information on research design is available in the Nature Research Reporting Summary linked to this article.

## Data availability

The NGS gene analysis data were deposited in the Sequence Read Archive (SRA) database (trace.ncbi.nlm.nih.gov), under the accession code SRP146080. Supplementary Figs. 3–5, Supplementary Figs. 6, 7, and Supplementary Figs. 10, 11 contain the uncropped gels corresponding to Fig. 4b, c, f, respectively. Source data underlying main figures are presented in Supplementary Data 1–6. All other data are available from the corresponding authors on reasonable request.

## Code availability

We used the following software in the analysis: G POWER 3.0, SHEsis (Bio-X Institute, SJTU), Image J 1.4, Compass Software 4.0 and Prism GraphPad 9.0. All code can be made available from the corresponding authors upon reasonable request.

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

## Acknowledgements

We are grateful to Analytical Chemistry platform, High-Throughput Screening platform, Cell Sorting and Imaging platforms, and Protein Expression and Preparation platform of Shanghai Institute for Advanced Immunochemical Studies, ShanghaiTech University for technical assistance with compound screen and flow cytometry experiments. We thank the staff members of the National Facility for Protein Science in Shanghai (NFPS), Zhangjiang Lab, China, for providing technical support and assistance in data collection, and analysis. This study was supported by Natural Science Foundation of Shanghai (20ZR1436800 to Yifeng Yang, 22ZR1442400 to Yan Zou), National Natural Science Foundation of China (No. 32170974 to Yan Zou, No. 31600686 to Jia Liu).

## Author contributions

Conceptualization, Y.F.Y. and L.H.; software, X.H.Z.; validation, W.W.L., X.H.Z., X.Y.W., X.Y.Y, Q.L.M, and Y.Y.Z.; formal analysis, Y.F.Y., M.N., and W.W.L.; data curation, X.H.Z., W.W.L, and M.N.; software, D. Z.; writing—original draft preparation, M.N. and Y.F.Y.; writing—review and editing J.L. and Y.Z.; supervision, B.J.; project administration, Y.F.Y. and Y.Z.; funding acquisition, J.L., Y.F.Y, and Y.Z. All authors have read and agreed to the published version of the manuscript.

## Competing interests

The authors declare no competing interests.
