## [Peer Review File · Communications Biology]

Reviewers' comments:

Reviewer #1 (Remarks to the Author):

This manuscript analyses a T/C polymorphism at rs895819 in the miR-27a gene that the authors identified as a mutation in patients with bipolar disorder (BD). The SNP was identified by screening a total of 1584 Han Chinese subjects using three groups, schizophrenia, BD, and healthy controls with roughly equal amounts of subjects. The effects of the T/C mutation on miR-27a function were analyzed in cell culture using U-251MG cells and human NPCs identifying that the T/C polymorphism led to reduced maturation of pre to mature miRNA. In addition, from 6 candidate target genes, the authors analyzed ICAM1 in more detail demonstrating that reduced expression of miR-27a revealed higher levels of ICAM1 protein.

This is a straightforward and largely well conducted study. The experimental set up is sound and the data are presented clearly. miRNAs have gained increasing interest in many diseases, including neuropsychiatric disorders but much of their functions are still not known. In the literature, there has been an increasing number of studies on miRNA/target relationships both on the descriptive and on the functional level. The strength of this investigation lies in the association of genetic variations identified in a larger cohort of subjects with functional analysis of a potential BD-associated mechanism, at least on the cellular level. Thus, this study adds to a growing literature on genetic predisposition in BD by identifying a novel T/C polymorphism of miR-27a and suggesting a hypothetical mechanism in disease pathology. The findings are somewhat consistent with previous data suggesting an association of increased ICAM1 levels with BD and the here described mechanism adds to further understanding the complexity of BD pathology.

Major:

To make a tighter case for miRNA function on ICAM1 regulation, in Fig. 5C, data on miR-27a WT and mutant should be shown in addition to mimics and inhibitors. Also, protein levels in Fig. 5C should be added and all experiments in Fig. 5 should be done in both U-251MG cells and NPCs.

The Discussion could be a bit sharper with regard to a role of ICAM1 in BD. For example, elevated ICAM1 levels are not specific to BD and have been found in other neuropsychiatric disorders and in aging as well. Also, there could be a bit more detailed discussion on the mechanisms of ICAM1 in disease pathogenesis. This can be at the expense of redundant text with the Result section.

Minor:

Lines 50-74: Please cite more original article(s) on miRNA biogenesis.

Line 109: To follow the flow of the text, Supplementary Table 2 should be Supplementary Table 1 and vice versa in the Methods section.

Fig. 1B: It should read "Circulatory system development"

Fig. 1D: It should read "Protein homodimerization activity"

Lines 331-351: There is quite a bit of redundant text with the Result section which should be avoided.

In my view, Fig. 6 is a mood point as it doesn't add much additional and clarifying information to this manuscript.

The English language is largely OK but needs corrections of some minor mistakes.

Reviewer #2 (Remarks to the Author):

The manuscript by Yang and colleagues has analyzed SNPs in microRNA genes in individuals with bipolar disorder, schizophrenia, and unrelated healthy people (~500 subjects each). The topic of the study is exciting and innovative as most studies so far have assessed microRNA expression levels in brain disorders, whereas much less have analyzed mutations in microRNA genes. They make the interesting discovery that a SNP in the gene coding for miR-27 is associated with bipolar disorder but not schizophrenia, and that this SNP leads to reduced expression of the mature

microRNA. Using RNAseq and qPCR in miR-27 KO astrocytoma cell lines and mimics and inhibitors in neuronal precursor cells they show differentially expressed mRNAs and identify ICAM-1 as a putative target that is regulated by miR-27. This is an interesting and novel finding, but the study could be strengthened in its rigor and significance. The major weakness is that it does not provide clear evidence for how the SNP effects neuronal function, which would be crucial to confirm its relevance for bipolar disorders.

Below are my specific comments.

Major comments:

- 1) The authors do not provide evidence for a causal relationship of the identified SNP and bipolar disorder, as neuronal phenotypes were not assessed. The paper would be strengthened considerably if the authors would provide some evidence for a neuronal dysfunction caused by the SNP.
- 2) I suggest combining table 2 and supplemental table 2 into one table. I think the association with bipolar disorder but not schizophrenia is particularly interesting.
- 3) The authors state that they identified two astrocytoma KO clones. The rigor of the study would be increased considerably if both were analyzed, at least using qRT-PCR to confirm RNASeq results.
- 4) The authors base their predicted target genes on mRNA expression. This approach does not consider the function of microRNAs as translational repressors. It is thus likely that many of the changes in gene expression that the scientists identified are secondary effects – in line of this assumption, more mRNAs were downregulated than upregulated in the KO cell lines. The authors should discuss this carefully.
- 5) Related to point 4, I think it would be helpful to analyze how many of the DEG are predicted targets of miR-27. Also, it would be helpful to show more direct evidence that ICAM is a target of miR-27, e.g. via luciferase assays using ICAM reporters along with ICAM reporter constructs with mutations in the seed sequence. If this data cannot be provided, I suggest toning down the statement that they identified ICAM as a target.
- 6) The authors find that the SNP, when expressed from plasmids in NPCs, significantly reduces microRNA expression. So, in my opinion the main finding is that the SNP reduces the levels of mature microRNA (which is interesting per se), and the reduced effects on target genes is then just secondary. It is thus not surprising that the SNP has a reduced effect on potential targets of miR-27. I thus suggest that this data (Fig. 4) should be shown before Fig. 3.
- 7) The quality of the western blots shown in 5A is low. Can the authors provide a different western blot and/or additional blots?

Other comments:

- 1) The authors state that they have not corrected for multiple testing in their studies but in figure legends and the methods they state that they did FDR. This should be clarified.
- 2) Statistics: Figure 3 should be analyzed as a 2-way ANOVA followed by posthoc and Fig. 4A should be a one-sample t-test to account for lack of variance in the WT.
- 3) In general, n=3 is low for these types of experiments.
- 4) The abstract seems to jump back and forth between bipolar disorders and cell lines, therefore suggest editing.
- 5) There are a few typos, e.g.: line 71: "interacts", line325: "to" validate

Response to reviewers

1) While we agree with Referee #2 that performing RNA-seq on the second miR27a-null astrocytoma clone would improve the impact of the study, any revision should, at a minimum, validate similar changes in gene expression between clonal lines using qRT-PCR. On a related note, we believe that any investigation of how rs895819 affects neuronal physiology would be interesting, but out of scope for the current manuscript, and should be discussed as a limitation.

Response: Thanks for your suggestion. We have reconstructed a **miR27a-null** neuroblastoma SH-SY5Y cell line and found the similar changes in gene expression, which were mostly significantly up-regulated in both miR-27a KO cell line by qPCR (see Fig 2).

We really agree to the importance of how rs895819 affects neuronal physiology. So we designed two functional experiments to find the possible link. One is to **detect neurotransmitter** expression of two stable NPCs containing mir-27a-C (mutant) or mir-27a-T(ctrl) , the other is focus on the **cell migration** because we identified NCAM1 as the target of miR-27a and NCAM1 is reported to have an effect on cell adhesion and cell migration. Indeed, we found that

mir-27a-C(mutant) affects both neurotransmitter expression and cell migration in neurons differentiated from NPCs.

In the manuscript we added one paragraph to describe this (Page 15, the first paragraph).

2.8 Up-regulation of NCAM1 by the Mir-27a-C Mutant suppresses cell migration and inhibits the secretion level of dopamine transmitter.

To evaluate the effects of rs895819 mutation on the neuron, we constructed mammalian expression with plasmids containing either WT (T) or mutant (C) miR-27a sequences (miR-27a-T and miR-27a-C) for overexpression. These experiments were performed in neural progenitor cells (NPCs). After 21 days of induction, the NPC cells were induced into neurons (Figure 6A). We found the dopamine transmitter was reduced in the miR-27a-C-Mutant, but the GABA was unchanged (Figure 6B,6C). Furthermore, cell migration was inhibited in the mir-27a-C-Mutant (Figure 6D,6E) in SH-SY5Y cell line. This is consistent with the dysregulation of the dopamine hypothesis in bipolar disorder.

We also added the description of these two methods on methods section (page 26).

(2) As suggested by Referee #2, examine a direct vs. indirect relationship between miR27a and ICAM, potentially using a 3' UTR luciferase assay. On a related note, we also agree with Referee #2 that the study would benefit from an analysis of which DEGs are predicted to harbor miR27a seed sequences or are potential targets.

Response: Good suggestion. According to the advice of Referee #2, we reanalyzed the DEGs which are predicted to harbor miR-27a seed sequence and acquired the top 20 gene list . (Table 2) and then did Luciferase Reporter Assay.

ICAM1 and NCAM1 were predicted as candidate targets of miR-27a by the verification of qPCR and western blotting.

To further examine the direct or indirect relationship between mir-27a and ICAM1 & NCAM1, we utilize 3' UTR luciferase reporter assay. We found relative luciferase activity of 3'UTR-NCAM1 was decreased dramatically in cells transfected with miR-27 mimic compared with miR-27 NC group, while up-regulation of miR-27 showed almost no effect on the relative luciferase activity of NCAM1-Mutant (Figure 5). This indicated that NCAM1 is a direct target as mir-27a.

(3) As noted by Referee #1, include WT and mutant miR27 as controls for the U-251MG experiments in Figure 5c, and quantify protein levels in all experimental conditions. If feasible, we also encourage you to repeat these experiments in NPCs, as a potentially relevant cell type for BD.

Response:

As suggested, we repeated all these experiments in NPCs . However, the results in NPC was not as effective in U251.This might be caused by low transfection efficiency of primary cells. Additionally, we tried many times and found that knocking out miR-27a will make NPC cells lose the ability to proliferate. On the other hand we successfully constructed the stable overexpression of miR-27a in NPCs. (page 23)

In addition, we have quantified protein levels in all experimental conditions.

(4) As suggested by both referees, please carefully proofread the manuscript for readability or grammatical errors.

Response: Thanks. We have searched for native speakers in helping paper editing for readability or grammatical errors.

Referee #1: miRNAs in neuropsychiatric disease and neuroinflammation

Referee #2: miRNAs in neurological disease

Reviewers' comments:

Reviewer #1 (Remarks to the Author):

This manuscript analyses a T/C polymorphism at rs895819 in the miR-27a gene that the authors identified as a mutation in patients with bipolar disorder (BD). The SNP was identified by screening a total of 1584 Han Chinese subjects using three groups, schizophrenia, BD, and healthy controls with roughly equal amounts of subjects. The effects of the T/C mutation on miR-27a function were analyzed in cell culture using U-251MG cells and human NPCs identifying that the T/C polymorphism led to reduced maturation of pre to mature miRNA. In addition, from 6 candidate target genes, the authors analyzed ICAM1 in more detail demonstrating that reduced expression of miR-27a revealed higher levels of ICAM1 protein.

This is a straightforward and largely well conducted study. The experimental set up is sound and the data are presented

clearly. miRNAs have gained increasing interest in many diseases, including neuropsychiatric disorders but much of their functions are still not known. In the literature, there has been an increasing number of studies on miRNA/target relationships both on the descriptive and on the functional level. The strength of this investigation lies in the association of genetic variations identified in a larger cohort of subjects with functional analysis of a potential BD-associated mechanism, at least on the cellular level. Thus, this study adds to a growing literature on genetic predisposition in BD by identifying a novel T/C polymorphism of miR-27a and suggesting a hypothetical mechanism in disease pathology. The findings are somewhat consistent with previous data suggesting an association of increased ICAM1 levels with BD and the here described mechanism adds to further understanding the complexity of BD pathology.

Response: We appreciate the positive comments on our paper.

Major:

To make a tighter case for miRNA function on ICAM1 regulation, in Fig. 5C, data on miR-27a WT and mutant should

be shown in addition to mimics and inhibitors. Also, protein levels in Fig. 5C should be added and all experiments in Fig. 5 should be done in both U-251MG cells and NPCs.

Response:

Thanks for your advice. Besides ICAM1, we also found another target gene NCAM1 for mir-27a using a new strategy from data analysis. According to your suggestion, we have done all of these experiments both in U251 and NPC.

The Discussion could be a bit sharper with regard to a role of ICAM1 in BD. For example, elevated ICAM1 levels are not specific to BD and have been found in other neuropsychiatric disorders and in aging as well. Also, there could be a bit more detailed discussion on the mechanisms of ICAM1 in disease pathogenesis. This can be at the expense of redundant text with the Result section.

Response: Because of the more evidence on NCAM1 than ICAM1 as the target of mir-27a, especially for the direct link between mir-27a and NCAM1 using Luciferase Reporter Assay, We rewritten the discussion section focus on the NCAM1 and

its role in BD.(see page 18-19)

Minor:

Lines 50-74: Please cite more original article(s) on miRNA biogenesis.

Response: As suggested, we have cited more original articles on miRNA biogenesis.

Referecne:

"Bartel DP. MicroRNAs: genomics, biogenesis, mechanism, and function. Cell. 2004 Jan 23;116(2):281-97.

Ambros V. The functions of animal microRNAs. Nature. 2004 Sep 16;431(7006):350-5.

Lewis BP, Burge CB, Bartel DP. Conserved seed pairing, often flanked by adenosines, indicates that thousands of human genes are microRNA targets. Cell. 2005 Jan 14;120(1):15-20. "

Line 109: To follow the flow of the text, Supplementary Table 2 should be Supplementary Table 1 and vice versa in the Methods section.

Response:OK, we have combined Table 2 and Supplementary Table 2 into one table, which was listed in advance along with

the associated method.

Fig. 1B: It should read "Circulatory system development"

OK, we have corrected this.

Fig. 1D: It should read "Protein homodimerization activity"

OK, we have corrected this.

Lines 331-351: There is quite a bit of redundant text with the Result section which should be avoided.

Thank you for pointing out this, and we have deleted this .

In my view, Fig. 6 is a mood point as it doesn't add much additional and clarifying information to this manuscript.

Response: We have replaced this figure with the functional data and outlined the possible mechanism of a functional SNP rs895819 on pre-miR-27a associated with bipolar disorder by targeting NCAM-1 in Figure 7.

The English language is largely OK but needs corrections of some minor mistakes.

Response: Thanks. We have asked for paper editing service to improve our manuscript.

Reviewer #2 (Remarks to the Author):

The manuscript by Yang and colleagues has analyzed SNPs in microRNA genes in individuals with bipolar disorder, schizophrenia, and unrelated healthy people (~500 subjects each). The topic of the study is exciting and innovative as most studies so far have assessed microRNA expression levels in brain disorders, whereas much less have analyzed mutations in microRNA genes. They make the interesting discovery that a SNP in the gene coding for miR-27 is associated with bipolar disorder but not schizophrenia, and that this SNP leads to reduced expression of the mature microRNA. Using RNAseq and qPCR in miR-27 KO astrocytoma cell lines and mimics and inhibitors in neuronal precursor cells they show differentially expressed mRNAs and identify ICAM-1 as a putative target that is regulated by miR-27. This is an interesting and novel finding, but the study could be strengthened in its rigor and significance. The major weakness is that it does not provide clear evidence for how the SNP effects neuronal function, which would be crucial to confirm its relevance for bipolar

disorders.

Response: Thanks a lot for your affirmation of our research innovation. To strength the function of SNP in bipolar disorder, we also consider it is very important to set up the link between SNP and neuronal function.

Below are my specific comments.

Major comments:

1) The authors do not provide evidence for a causal relationship of the identified SNP and bipolar disorder, as neuronal phenotypes were not assessed. The paper would be strengthened considerably if the authors would provide some evidence for a neuronal dysfunction caused by the SNP.

Response:

Thanks for your precious advice. We agree to this .

So we have constructed the stable mir-27a overexpression cell line of NPC and then induced NPC into neuron to detect the neurotransmitter level by ELISA method. We found the dopamine transmitter was reduced in miR-27a-C-Mutant. This is consistent with dysregulation of dopamine hypothesis in bipolar disorder. We added this in Page 15, the last paragraph.

Another link evidence is NCAM1. We identified NCAM1 as the direct target of mir27a through qPCR, western blotting and luciferase reporter assay. As we noted in the DISCUSSION section: The neural cell adhesion molecule (NCAM) belongs to the immunoglobulin superfamily and is a cell recognition molecule bound to the membrane. It is involved in the development of the nervous system by regulating synaptic plasticity, neurite outgrowth, neuronal migration, and synapse formation. Different NCAM functions are related to psychiatric disorders. Additionally, single nucleotide polymorphisms found in the NCAM gene, the abnormal proteolysis or polysialylation of the NCAM protein can change its role in diseases related to neuropsychiatry. All these indicates that NCAM might play an important role in bipolar disorder.

(Page 18)

2) I suggest combining table 2 and supplemental table 2 into one table. I think the association with bipolar disorder but not

schizophrenia is particularly interesting.

Response: We have combined these two table into ONE and discussed the difference in the discussion section.

3) The authors state that they identified two astrocytoma KO clones. The rigor of the study would be increased considerably if both were analyzed, at least using qRT-PCR to confirm RNASeq results.

Response: Yes, we have identified two astrocytoma KO clones one in U251 and the other in SH-SY5Y.

4) The authors base their predicted target genes on mRNA expression. This approach does not consider the function of microRNAs as translational repressors. It is thus likely that many of the changes in gene expression that the scientists identified are secondary effects – in line of this assumption, more mRNAs were downregulated than upregulated in the KO cell lines. The authors should discuss this carefully.

Response: Thanks for your point. To minimize the potential secondary effects, we reconstructed three U251 KO using 3 different sgRNAs, and verified several upregulated genes including ICAM1 and NCAM1 as candidate targets. Then we used 3'UTR luciferase reporter assay to confirm that NCAM1

3'UTR can be directly targeted by mir-27a, leading to a reduction in NCAM1 protein levels.

5) Related to point 4, I think it would be helpful to analyze how many of the DEG are predicted targets of miR-27. Also, it would be helpful to show more direct evidence that ICAM is a target of miR-27, e.g. via luciferase assays using ICAM reporters along with ICAM reporter constructs with mutations in the seed sequence. If this data cannot be provided, I suggest toning down the statement that they identified ICAM as a target.

Response: Thank you for the suggestions. We have fused 3'UTR sequences (WT vs. seed region mutants) of candidate genes with luciferase as reports to verify whether these gene are direct targeted by mir-27a (see Fig 5). Interestingly, NCAM1 but not ICAM1 is a direct target of mir-27a.

6) The authors find that the SNP, when expressed from plasmids in NPCs, significantly reduces microRNA expression. So, in my opinion the main finding is that the SNP reduces the levels of mature microRNA (which is interesting per se), and the reduced effects on target genes is then just secondary. It is

thus not surprising that the SNP has a reduced effect on potential targets of miR-27. I thus suggest that this data (Fig. 4) should be shown before Fig. 3.

Response: We agree to this. We have shown the “SNP reduces the levels of mature microRNA ” data in advance. What is more, to investigate the possible SNP function in the pathology of bipolar disorder, we designed the function analysis to detect neurotransmitter level instead of the reduced effect of potential targets of miR27. We think it is redundant information because we have identified NCAM1 as direct target as miR-27a , and NCAM1 was reported to have an important role in psychiatry diseases including Bipolar disorder.

7) The quality of the western blots shown in 5A is low. Can the authors provide a different western blot and/or additional blots?

Response: To improve our data quality, we utilize JESS (Protein Simple Company) to do most of our protein quantification.

Other comments:

1) The authors state that they have not corrected for multiple

testing in their studies but in figure legends and the methods they state that they did FDR. This should be clarified.

Response: Here "Multiple testing" specially denotes Bonferroni test in the population research. To avoid misunderstanding, we added "perform in the population "

2) Statistics: Figure 3 should be analyzed as a 2-way ANOVA followed by posthoc and Fig. 4A should be a one-sample t-test to account for lack of variance in the WT.

Response: Thanks. We have reanalyzed the data according to your suggestions.

3) In general, $n=3$ is low for these types of experiments.

Response: As you suggested, it is low for $n=3$. Fortunately, we repeated it several times and also used different methods to verify.

4) The abstract seems to jump back and forth between bipolar disorders and cell lines, therefore suggest editing.

Response: Thanks. We have revised the abstract.

5) There are a few typos, e.g.: line 71: "interacts", line325: "to" validate

Response: Thanks for you correction. We have corrected them.

Reviewers' comments:

Reviewer #1 (Remarks to the Author):

The authors have sufficiently addressed the reviewers' points and added additional data to improve the quality and message of the manuscript. There are a few minor revisions with regards to my original comments:

-Please review the legend to revised Figure 4. It should be clear which data are mRNA and which are protein.

-The observation that k.o. of miR-27a in NPCs negatively affects their proliferation is interesting and may warrant a bit more attention. With regards to cell function, there seems to be a difference between knocking out miR-27a and reducing its maturation via the C polymorphism. Do the miR-27a mutant NPCs also have reduced cell proliferation? In my view, discussing the mechanistic difference between gene k.o. versus polymorphism-mediated regulation of miRNA maturation is an important point, particularly in context of cell type-specific functions, e.g., cell proliferation in NPC, as an important factor in brain development. I suggest to adding a few sentences to address this issue in more detail.

Reviewer #2 (Remarks to the Author):

In response to the previous comments, the authors added new data, which, in theory, addresses the concerns. However, many details are missing from results and legend sections, the description of the results is scarce and even misleading in some cases, and for some, the number of repeats and statistical tests are not indicated. While western blots are quantified now, no bar diagrams with error bars to illustrate variability, nor statistics are provided. Some of the data are inconsistent (effects of the three sgRNAs in 4C,D) but not discussed, and there seems to be a mix up both in legend and quantification.

1) The new data in Figure 4 are interesting but somewhat confusing and need clarification and a more accurate description.

a. It seems as if the mimic and/or inhibitor controls have an effect themselves, in particular in the ICAM1 experiments. In both cell lines, ICAM levels in cells treated with the negative control inhibitor are strongly reduced compared to in those cells treated with the mimic negative control. This strongly confounds the experiment and should be explained.

b. The left two blots in B and the blots in D look as if they were generated using a Jess system whereas the two right blots in B look like traditional western blots. This should be specified. More importantly, quantifications should be shown.

c. The legend is incorrect (B shows western blots but is described as qRT data) and lacks a lot of detail.

d. The results section states that three different KO cells generated with three different sgRNAs increased NCAM1 expression; however, only 1 of 3 increased the mRNA, and ICAM1 was unchanged, that needs to be better described and discussed. D only showed one example and the quantification is confusing (see point f).

e. The authors list values below their western blots – it is unclear if these are the averages of the three replicates (if there were three replicates, that is also unclear from the legend) or just the quantification of the single experiment. The authors should show bar diagrams with individual dots for each replication throughout the figure.

f. The quantification in D does not match what is seen in the blot (e.g., value 0 for ICAM1 although there is clearly a band). They might have mixed up NCAM1 with ICAM1.

2) For figure 5, the authors should better describe which nucleotides were mutated in their "mutant control".

3) In the description of figure 6, the authors should better explain that they analyzed the cell culture media. In addition, the transwell assay should be better described so that the reader can assess what it shows. The legend does not indicate repeats, number of cultures or statistics.

4) Figure 7 (model): it should make clear that the last sentences about the role of NCAM1 in dopamine release and cell migration is not shown in this study and rather speculative.

- 5) The statistics section should be more specific about what type of tests were used for which data, and all legends should indicate the specific test as well as type of posthoc. When 2-way ANOVA was done, the statistics should be reported.
- 6) Sequences of antagomirs, mimics and sg RNAs (in addition to the mutation in NCAM1 as indicated above) should be listed.

Referee expertise:

Referee #1: miRNAs in neuropsychiatric disease and neuroinflammation

Referee #2: miRNAs in neurological disease

Reviewers' comments:

Reviewer #1 (Remarks to the Author):

The authors have sufficiently addressed the reviewers' points and added additional data to improve the quality and message of the manuscript.

Response: Thanks for your positive comments for our revised manuscript.

There are a few minor revisions with regards to my original comments:

Please review the legend to revised Figure 4. It should be clear which data are mRNA and which are protein.

Response : Thanks for your correction. We have corrected this in Figure 4 legend.

In our former revision, we used two quantification methods of protein expression including traditional western blotting (semi-quantitative) and Jess

(automatic protein expression quantitative analysis system, see SUPPLEMENTARY method. Taking into account good repeat-ability, more accurate quantitative data, and data consistency, we have repeated all the experiments of protein quantification at least 3 times using Jess system.

The observation that k.o. of miR-27a in NPCs negatively affects their proliferation is interesting and may warrant a bit more attention. With regards to cell function, there seems to be a difference between knocking out miR-27a and reducing its maturation via the C polymorphism.

Response: Yes. It's really interesting that KO of miR-27a in NPCs affects their proliferation. In our study, the rs895919 mutation on pre-miR-27a affects the maturation of miR-27a and inhibits the miR-27 expression in NPC. To discriminate the difference between CRISPR/Cas9-mediated knocking out miR-27a and reducing its maturation via the C polymorphism, we investigated the WT, knockout, C allele, T allele of miR-27a, and found that miR-27 KO inhibits the cell growth in SH-SY5Y at 72 hours using CCK8 method. No significant difference was observed in the cell growth between C-allele and T-allele. (supp Fig. 5)

SUPPLEMENTARY Figure 5. The knockout of miR-27a inhibits the growth of SH-SY5Y.

Proliferation of several cell lines (WT, C allele, T allele and miR-27a KO in SH-SY5Y) was examined using the CCK8 assay at 24, 48 and 72 hours, respectively. Significant difference between WT and KO were analyzed by one-way ANOVA; n=5 (*, $P < 0.05$; **, $P < 0.01$; ***, $P < 0.001$; n.s., not significant).

Do the miR-27a mutant NPCs also have reduced cell proliferation?

Response: In order to answer this question, we used CCK8 method to detect the OD450 values of NPC-C allele / NPC-T allele at 48 hours and we found no significance between them. In summary, the rs895919 mutation on pre-miR-27a affects the maturation of miR-27a but has no effect on the cell growth rate in NPCs. (in supplementary Fig 4)

Supplementary Figure 4. The C mutant has no effect on NPC proliferation.

Proliferation of NPC Ctrl (T allele) and NPC Mutant (C allele) was examined using the CCK8 assay at 48 hours. Unpaired t-test was performed between two groups with five replicates (n=5) for each group. Results were expressed as the mean ± SEM. P<0.05 as a sign of significance; n.s., not significant.

In my view, discussing the mechanistic difference between gene k.o. versus polymorphism-mediated regulation of miRNA maturation is an important point, particularly in context of cell type-specific functions, e.g., cell proliferation in NPC, as an important factor in brain development. I suggest to adding a few sentences to address this issue in more detail.

Response: Thanks for your good suggestion. “miR-27a knockout cell line and the miR-27a point mutation cell line are different in the cell physiological character and its functional mechanism. In our study, we found that nucleotide polymorphisms (SNPs) or mutations occurring in the miRNA gene region may

affect the property of miRNAs through altering miRNA expression and/or maturation (see Fig. 3). On the other hand, miR-27a KO in NPCs affects their proliferation (see FigS4) . Meanwhile, the cell growth is inhibited in miR-27a knockout of SH-SY5Y. CRISPR/Cas9-mediated knockout of all miR-27/24 in ESCs leads to serious deficiency in ESC differentiation in vitro and in vivo [1]. Therefore, we think that miR-27a also plays an important role in the proliferation and differentiation of NPC, and its function can be further studied in future experiments.”

We added this above paragraph in the discussion section (Line 285-296).

Reviewer #2 (Remarks to the Author):

In response to the previous comments, the authors added new data, which, in theory, addresses the concerns. However, many details are missing from results and legend sections, the description of the results is scarce and even misleading in some cases, and for some, the number of repeats and statistical tests are not indicated. While western blots are quantified now, no bar diagrams with error bars to illustrate variability, nor statistics are provided. Some of the data are inconsistent (effects of the three sgRNAs in 4C,D) but not discussed, and there seems to be a mix up both in legend and

quantification.

Response: Thanks a lot for your efforts to improve the quality of our data. We have provided more details for the data in the figure legend including the number of repeats, specific statistics for every experiment and bar diagrams with error bars to illustrate variability. (Figure 2, line 157-158; Figure 3, line 182-183; Figure 4, line 206-218; Figure 5, line 234-235; Figure 6, line 255, 257-259).

1) The new data in Figure 4 are interesting but somewhat confusing and need clarification and a more accurate description.

a. It seems as if the mimic and/or inhibitor controls have an effect themselves, in particular in the ICAM1 experiments. In both cell lines, ICAM levels in cells treated with the negative control inhibitor are strongly reduced compared to in those cells treated with the mimic negative control. This strongly confounds the experiment and should be explained.

Response: We analyzed the possible reasons for the irregular phenomena caused by the mimic/inhibitor controls. The addition of transfection reagent and RNA might be toxic to cells, resulting in changes in gene expression. Firstly, we replaced Lipofectamine 3000 with RNAimax (Invitrogen, which is more effective for RNA transfection and less toxic than lipofectamine 3000). We replaced another Negative control for RNA sequence and reduced its dosage. Especially in NPC, we used 1 pmol or 10 pmol mimic or inhibitor RNA in total. Then we evaluated the effect of miR-27a mimics or inhibitor by qPCR

method. The relative expression of mature miR-27a was dramatically increased by the miR-27a mimic and decreased by the miR-27a inhibitor ($p < 0.0001$) in U251 and NPC (Figure 4A), which showed the effectiveness of miR-27a mimics and inhibitors (Line 195-203). In addition, the negative control changes little compared to wild types after optimization.

b. The left two blots in B and the blots in D look as if they were generated using a Jess system whereas the two right blots in B look like traditional western blots. This should be specified. More importantly, quantifications should be shown.

Response: We agree with you. Taking into account Jess's good repeat-ability, more accurate quantitative data, and data consistency, we have repeated this part of the experiments using Jess system with at least three replicates.

c. The legend is incorrect (B shows western blots but is described as qRT data) and lacks a lot of detail.

Response: We are very sorry for our incorrect description of legend to figure 4B. In addition, we have added experimental information in detail according to reviewer's suggestion.

d. The results section states that three different KO cells generated with three different sgRNAs increased NCAM1 expression; however, only 1 of 3 increased the mRNA, and ICAM1 was unchanged, that needs to be better described and discussed. D only showed one example and the quantification is confusing (see point f).

Response: To avoid the off-target of CRISPR, three KO cells were generated with three different sgRNAs. These three sgRNAs effectively inhibited the mature of miR-27a (Fig 4D). In addition, we found that in those KO cell lines the protein levels of both NCAM1 and ICAM1 are increased (Fig. 4F), but are not necessarily consistent with their mRNA levels (Figure 4E).

The mRNA abundance of a specific gene may not have a linear relationship with the expression level of its translation product-protein, because there are many levels of regulation of gene expression, and the regulation of transcription level is only one link. There are also post-transcriptional regulation and translational and post-translational regulation. Furthermore, factors such as mRNA degradation, protein degradation, and modified folding may cause the abundance of mRNA to be inconsistent with the level of protein expression.

Finally, we think that the amount of protein expression plays a major role in cellular functions and thus NCAM1 and ICAM1 expression are up-regulated in the three miR-27a KO cell lines.

e. The authors list values below their western blots – it is unclear if these are the averages of the three replicates (if there were three replicates, that is also unclear from the legend) or just the quantification of the single experiment. The authors should show bar diagrams with individual dots for each replication throughout the figure.

Response: According to reviewer's suggestion, we have showed bar diagrams

for the replications throughout the figures.

f. The quantification in D does not match what is seen in the blot (e.g., value 0 for ICAM1 although there is clearly a band). They might have mixed up NCAM1 with ICAM1.

Response: We are very sorry for negligence of this. We have corrected it and provided more details in it.

2) For figure 5, the authors should better describe which nucleotides were mutated in their “mutant control”.

Response: Thanks for your valuable suggestion. We have provided the mutated information in **Fig 5A**.

3) In the description of figure 6, the authors should better explain that they analyzed the cell culture media. In addition, the transwell assay should be better described so that the reader can assess what it shows. The legend does not indicate repeats, number of cultures or statistics.

Response: To give a better understanding of Figure 6, we described the following paragraph.

“The normal release of neurotransmitters plays an important role in maintaining the normal physiological functions of the body. Abnormalities in the release and regulation of neurotransmitters are also related to a series of pathological processes, such as depression, bipolar disorder and Alzheimer's disease (AD) [2-5]. There are hundreds of neurotransmitters in the human brain, including both excitatory neurotransmitters and inhibitory

neurotransmitters. Dopamine (one of excitatory neurotransmitters) and GABA (one of inhibitory neurotransmitters) have been reported to play important roles in bipolar disorder. (Line 313-320)

After 21 day induction, we successfully induced the NPCs (nestin⁺sox⁺) into neurons (MAP2⁺ TUJ1⁺). To investigate the release of neurotransmitters, we collected the cell culture of two kinds of neurons (induction individually from NPC-C-allele and NPC-T-allele) and then detected the secretory neurotransmitters, dopamine and GABA, using ELISA method." (Line 506-509)

As for transwell assay, we also described more details in methods.(Line 512, 514-520)

"Transwell chamber with 8µm pore size polycarbonate filter inserts for 24-well plates (Corning Costar Corporation, Cambridge, MA) were used to examine the migration of SH-SY5Y cells. We used a layer of membrane (matrigel, Biocoat, 354480) to separate the high-nutrient culture solution from the low-nutrient culture solution. First, 100µl 5% matrigel (diluted by DMEM) was spread in the upper chamber and incubated at 37°C for 2 hours. Then we used the tips to suck off the excess matrigel and washed it again with PBS. Next, 1x10⁴ cells were seeded into the upper chamber separately with 200µl serum (free DMEM). At the same time, 500µl DMEM containing 20% FBS was added into the lower chamber."

4) Figure 7 (model): it should make clear that the last sentences about the role of NCAM1 in dopamine release and cell migration is not shown in this study and rather speculative.

Response: Thanks for the suggestion. We have added the following paragraph:

“It is hypothesized that up-regulation of NCAM1 might suppress the cell migration and dopamine expression levels and thus result in the dysregulation.”

(line 344-345)

5) The statistics section should be more specific about what type of tests were used for which data, and all legends should indicate the specific test as well as type of posthoc. When 2-way ANOVA was done, the statistics should be reported.

Response: According to your suggestion, we have reported the specific statistics for our experiments. When 2-way ANOVA was done, we have reported the statistics in the legend of Fig 3B.

6) Sequences of antagomiRs, mimics and sg RNAs (in addition to the mutation in NCAM1 as indicated above) should be listed.

Response: As you suggested, sequence of antagomiRs, mimics and sgRNAs have been listed in the supplementary Table.

Reference

1. Ma Y, Yao N, Liu G, Dong L, Liu Y, Zhang M, Wang F, Wang B, Wei X, Dong H, Wang L, Ji S, Zhang J, Wang Y, Huang Y, Yu J. Functional screen reveals essential roles of miR-27a/24 in differentiation of embryonic stem cells. *EMBO J*. 2015 Feb 3;34(3):361-78. doi: 10.15252/embj.201489957. Epub 2014 Dec 17. PMID: 25519956
2. Rosenblatt S, Leighton WP, Chanley JD. Dopamine-beta-hydroxylase: evidence for increased activity in sympathetic neurons during psychotic states. *Science*. 1973 Nov 20;182(4115):923-4. doi: 10.1126/science.182.4115.923. PMID: 4745595.
3. Ashok AH, Marques TR, Jauhar S, Nour MM, Goodwin GM, Young AH, Howes OD. The dopamine hypothesis of bipolar affective disorder: the state of the art and implications for treatment. *Mol Psychiatry*. 2017 May;22(5):666-679. doi: 10.1038/mp.2017.16. Epub 2017 Mar 14. PMID: 28289283
4. Lener MS, Niciu MJ, Ballard ED, Park M, Park LT, Nugent AC, Zarate CA Jr. Glutamate and Gamma-Aminobutyric Acid Systems in the Pathophysiology of Major Depression and Antidepressant Response to Ketamine. *Biol Psychiatry*. 2017 May 15;81(10):886-897. doi: 10.1016/j.biopsych.2016.05.005. Epub 2016 May 12. PMID: 27449797
5. Brady RO Jr, McCarthy JM, Prescott AP, Jensen JE, Cooper AJ, Cohen BM, Renshaw PF, Ongür D. Brain gamma-aminobutyric acid (GABA) abnormalities

in bipolar disorder. *Bipolar Disord.* 2013 Jun;15(4):434-9. doi:
10.1111/bdi.12074. Epub 2013 May 2. PMID: 23634979

REVIEWERS' COMMENTS:

Reviewer #1 (Remarks to the Author):

The authors have improved the quality of their data and straightened out the remaining issues. There are a couple of issues with the English language, which probably can be resolved when processing the proofs.

Reviewer #2 (Remarks to the Author):

The authors have addressed most of my comments sufficiently. There are still a few things that should be added or changed.

- 1) Figure 3: For two-ANOVA statistics, it would be important what p value is reported (one of the main effects or the interaction?). This information should be added.
- 2) Figure 4: The new quantification in G should indicate the statistical test used. It would also be helpful to indicate that G is the quantification of NCAM1 and ICAM1 protein levels in U251 cells. D should indicate in the y axis label that miR-27 was quantified.
- 3) Line 506: This should read "the cell culture supernatant" or "...cell culture media"

Figure 3: For two-ANOVA statistics, it would be important what p value is reported (one of the main effects or the interaction?). This information should be added.	Response : As you suggested, we have reported that p value is reported to one of the main effects for two-way ANOVA statistics . (Line 554-555, Page 22)
Figure 4: The new quantification in G should indicate the statistical test used. It would also be helpful to indicate that G is the quantification of NCAM1 and ICAM1 protein levels in U251 cells. D should indicate in the y axis label that miR-27 was quantified.	Response:  1.We have added the the statistical test used in Figure 4 G. (Line 570-572,Page 23) 2.We also indicated that G is the quantification of NCAM1 and ICAM1 protein levels in U251 cells.(Line 570,Page 23). 3. Figure 4D indicated in the y axis label that miR-27 was quantified.
Line 506: This should read “the cell culture supernatant” or “...cell culture media”	Response: Thanks for your correction. We have corrected it as “the cell culture media” in the new revision. (Line 441, Page 16)